# Increased TGF-β and BMP Levels and Improved Chondrocyte-Specific Marker Expression In Vitro under Cartilage-Specific Physiological Osmolarity

**DOI:** 10.3390/ijms20040795

**Published:** 2019-02-13

**Authors:** Ufuk Tan Timur, Marjolein Caron, Guus van den Akker, Anna van der Windt, Jenny Visser, Lodewijk van Rhijn, Harrie Weinans, Tim Welting, Pieter Emans, Holger Jahr

**Affiliations:** 1Laboratory for Experimental Orthopedics, Department of Orthopaedic Surgery, Maastricht University Medical Centre+, 6229 HX Maastricht, The Netherlands; marjolein.caron@maastrichtuniversity.nl (M.C.); g.vandenakker@maastrichtuniversity.nl (G.v.d.A.); L.van.rhijn@mumc.nl (L.v.R.); t.welting@maastrichtuniversity.nl (T.W.); p.emans@mumc.nl (P.E.); hjahr@ukaachen.de (H.J.); 2Institute of Anatomy and Cell Biology, RWTH Aachen University, 52074 Aachen, Germany.; 3Department of Orthopaedics, Erasmus MC, University Medical Center, 3000 CA Rotterdam, The Netherlands; a.vanderwindt@erasmusmc.nl; 4Department of Internal Medicine, Erasmus MC, University Medical Center, 3000 CA Rotterdam, The Netherlands; j.visser@erasmusmc.nl; 5Department of Orthopaedics, University Medical Center Utrecht, 3508 GA Utrecht, The Netherlands; h.h.weinans@umcutrecht.nl; 6Department of Rheumatology & Clinical Immunology, University Medical Center Utrecht, 3508 GA Utrecht, The Netherlands; 7Department of Biomechanical Engineering, Delft University of Technology, 2628 CD Delft, The Netherlands

**Keywords:** chondrocyte, osmolarity, TGF-β superfamily, signalling, collagen type II, bone morphogenetic proteins

## Abstract

During standard expansion culture (i.e., plasma osmolarity, 280 mOsm) human articular chondrocytes dedifferentiate, making them inappropriate for autologous chondrocyte implantation to treat cartilage defects. Increasing the osmolarity of culture media to physiological osmolarity levels of cartilage (i.e., 380 mOsm), increases collagen type II (*COL2A1*) expression of human articular chondrocytes *in vitro*, but the underlying molecular mechanism is not fully understood. We hypothesized that TGF-β superfamily signaling may drive expression of *COL2A1* under physiological osmolarity culture conditions. Human articular chondrocytes were cultured in cytokine-free medium of 280 or 380 mOsm with or without siRNA mediated TGF-β2 knockdown (RNAi). Expression of TGF-β isoforms, and collagen type II was evaluated by RT-qPCR and immunoblotting. TGF-β2 protein secretion was evaluated using ELISA and TGF-β bioactivity was determined using an established reporter assay. Involvement of BMP signaling was investigated by culturing human articular chondrocytes in the presence or absence of BMP inhibitor dorsomorphin and BMP bioactivity was determined using an established reporter assay. Physiological cartilage osmolarity (i.e., physosmolarity) most prominently increased TGF-β2 mRNA expression and protein secretion as well as TGF-β bioactivity. Upon TGF-β2 isoform-specific knockdown, gene expression of chondrocyte marker *COL2A1* was induced. TGF-β2 RNAi under physosmolarity enhanced TGF-β bioactivity. BMP bioactivity increased upon physosmotic treatment, but was not related to TGF-β2 RNAi. In contrast, dorsomorphin inhibited *COL2A1* mRNA expression in human articular chondrocytes independent of the osmotic condition. Our data suggest a role for TGF-β superfamily member signaling in physosmolarity-induced mRNA expression of collagen type II. As physosmotic conditions favor the expression of *COL2A1* independent of our manipulations, contribution of other metabolic, post-transcriptional or epigenetic factors cannot be excluded in the underlying complex and interdependent regulation of marker gene expression. Dissecting these molecular mechanisms holds potential to further improve future cell-based chondral repair strategies.

## 1. Introduction:

Articular cartilage defects do not heal spontaneously and recent data suggest that treatment of these cartilage defects by microfracture (MF) procedures are inferior to autologous chondrocyte implantation (ACI) [1,2] due to biomechanically inferior repair tissue [3]. Despite ACI being superior to microfracture, it is not always characterized by hyaline cartilage repair tissue [4]. Further improvements of currently available chondrocyte-based treatment approaches such as the ACI are thus needed.

Chondrocytes are the main cell type found in cartilage (i.e., about 1–5% *v*/*v*) and responsible for maintaining the cartilage extracellular matrix (ECM) [5]. ACI involves in vitro expansion of human articular chondrocytes (HACs) prior to re-implantation into the cartilage defect. During in vitro expansion chondrocytes inevitably lose their specific phenotype and de-differentiate with increasing passages making them gradually less suitable for autologous cell therapies [6]. Dedifferentiation of HACs is characterized by reduced expression of aggrecan (ACAN) and collagen type II (*COL2A1) mRNA*, probably being the most important cartilage-specific markers, and accompanied by an increased fibroblast-like phenotype [6].

Cartilage requires a high osmotic value of its interstitial fluid to maintain its hydrostatic pressure and viscoelastic properties, for which negatively charged glycosaminoglycan side chains of the proteoglycans (PGs) are crucial to attract mobile cations and water into the ECM environment. The intact collagen network, in contrast, restricts the inherent swelling force of the ECM and determines the relatively high osmolarity of this extracellular fluid [5]. The latter ranges between 380 and 450 mOsm in healthy cartilage (i.e., physiological osmolarity) [5], which is markedly higher than that of plasma levels or standard culture medium (around 280 mOsm) [7]. In analogy to using “physoxia” to describe physiological oxygenation levels in cartilage, avoiding the term hypoxia [8], we would like to introduce physiological osmolarity as “physosmotic”. We and others have already shown that cultured chondrocytes are osmo-responsive and improve ECM synthesis accordingly [5,9,10]. However, the underlying molecular mechanism is poorly understood.

Transforming growth factor (TGF-β) plays an indispensable role in cartilage repair and homeostasis [11,12]. This pleiotropic growth factor is the prototypic member of the TGF-β superfamily, which further includes, among others, the large subfamily of bone morphogenetic proteins (BMPs) [13]. Three mammalian isoforms of TGF-β have been described, with non-redundant functions in vivo [14,15]. In chondrocytes, TGF-β has been implicated in the transcription of collagen type II [16]. Canonically, TGF-β ligands bind to the TGF-β type II receptor (TGFBRII) which, in most cells, then recruits the type I receptor activin-like kinase 5 (ALK5) to phosphorylate intracellular effector molecules SMAD-2 and -3 to regulate transcriptional activity [17]. It was shown that in chondrocytes TGF-β can also signal through an alternative receptor, ALK1 [18] to phosphorylate SMAD1/5/8, rather than SMAD2/3. The ALK5/ALK1 ratio now appears to be responsible for the seemingly contradictory actions of TGF-β in chondrocytes [19]. Context-dependent ambivalent actions of TGF-β on articular chondrocytes have been reviewed [12], underscoring its key role in chondrocyte ECM synthesis and homeostasis. 

Since TGF-β is known to induce *COL2A1* expression in chondrocytes [18], it is tempting to speculate that increased *COL2A1* expression in in vitro HAC cultures upon physosmotic treatment, may be caused by stimulated TGF-β signaling. Currently, however, little is known about how osmolarity may induce endogenous TGF-β signalling, which holds especially for chondrocytes. Dissecting molecular mechanisms underlying physosmotic induction of chondrocyte markers might aid in further improving cell-based chondral repair strategies, as well as improving HAC culture conditions for research purposes [20].

In the present study, we therefore aim to elucidate whether changes in TGF-β signalling underlie the cartilage physosmolar induction of chondrocyte marker gene expression in in vitro HAC cultures.

## 2. Results

### 2.1. Cartilage Physosmotic Culture Induces Specific TGF-β Isoform Expression in HACs 

Confirming our earlier results [10], culturing HACs for seven days in physosmotic conditions (380 mOsm, physosmotic medium, PM) results in significantly (*p* ≤ 0.0001) elevated *COL2A1* mRNA expression, as compared to 280 mOsm (i.e., osmotic control medium, OCM) (Figure 1A). To determine whether this increased COL2A1 expression upon physosmotic treatment may be caused by stimulated specific TGF-β isoform induced signalling, mRNA expression of the three human TGF-β isoforms (*TGFB1-3*) was measured. PM selectively induced expression of *TGFB2* (two-fold, *p* < 0.0001)) and *TGFB3* (1.5-fold, *p* < 0.0001), respectively. In contrast, expression of *TGFB1* was not significantly altered by changes in medium osmolarity (Figure 1B).

### 2.2. Physosmolarity Increases Secretion of Bioactive TGF-β2 

As PM most prominently increased *TGFB2* mRNA abundance, we next aimed at confirming if this resulted in increased TGF-β2 protein level, using a TGF-β2 isoform-specific ELISA assay. HACs were cultured in either serum free (SF) medium or with 10% FCS and TGF-β2 secretion in culture supernatants was measured after seven days. TGF-β2 secretion was largely independent of the presence of serum in the culture medium and showed a 1.6-fold (10% FCS, *p* = 0.003) and two-fold (serum-free, *p* = 0.006) increase in PM as compared to OCM (Figure 2A). The osmolarity-dependent change in *TGFB2* gene expression is thus predictive for TGF-β2 protein secretion in culture supernatant in response to PM. To determine whether the secreted TGF-β2 is also bioactive, we performed an established bioassay that reports a TGF-β specific activation of the TGF-β signaling pathway. Briefly, the TGF-β-responsive SMAD response element-mediated firefly luciferase signal is normalized to a constitutively expressed Renilla luciferase signal, to provide relative activity [21]. In agreement with increased TGF-β2 secretion in PM, this conditioned PM had a significantly (*p* < 0.0001) increased TGF-β bioactivity (2.3-fold; Figure 2B), as compared to conditioned OCM. 

On the receptor level ALK5, but not ALK1, appears to be essential for TGF-β induced phosphorylation of SMAD3 [18], and therefore we further investigated the expression ratio of *ALK5*/*ALK1* in response to PM to get an indication if the increased TGF-β might be able to induce canonical signaling. In accordance with the TGF-β bioassay, the *ALK5/ALK1* expression ratio was significantly shifted in favor of increased ALK5 expression in chondrocytes cultured in PM as compared to OCM (*p* = 0.027) (Figure 2C). These data indicate that physosmotic culture conditions improve activation of ALK5 driven TGF-β signaling.

### 2.3. TGF-β2 Isoform-Specific Knockdown in HAC Cultures In Vitro 

To investigate whether a causal relation exists between the increased *COL2A1* expression and TGF-β2 secretion after treatment with PM, we performed TGF-β2 knockdown experiments using RNAi (siRNA) in our HAC cultures. First, we evaluated the isoform-specificity of RNAi before treatment with PM at t = 0: TGF-β2 gene expression was specifically and significantly down-regulated upon RNAi when compared to control and other TGF-β isoforms (Figure 3) (*p* = 0.001). 

### 2.4. TGF-β2 RNAi Combined with Physosmotic Treatment Increases COL2A1 Gene Expression in HACs 

We subsequently determined the consequence of TGF-β2 RNAi on *COL2A1* gene expression 48 h upon stimulation with PM as compared to OCM. PM significantly increased *COL2A1* after 48 h (i.e., 1.8-fold, *p* = 0.01) (Figure 4A). 

TGF-β2 knockdown, did not alter *COL2A1* expression in OCM (Figure 4A). However, TGF-β2 RNAi with PM stimulation after 48 h resulted in higher *COL2A1* expression compared to OA HACs cultured at PM without TGF-β2 RNAi (*p* = 0.04) (Figure 4A). 

To determine if the increased *COL2A1* expression upon TGF-β2 knockdown under physosmotic conditions (i.e., in PM) was accompanied by an increased reporter assay activity, we repeated the experiment in the established chondrosarcoma cell line SW1353 [22,23]. We measured CAGA-12 luciferase activity 48 h after stimulation with PM with TGF-β2 RNAi. Bioactivity of TGF-β displayed a similar pattern as *COL2A1* expression: stimulation with PM increased CAGA-12 luciferase activity (*p* = 0.03) as compared to the control condition (i.e., OCM, Figure 4B). While TGF-β2 knockdown at OCM did not alter CAGA-12 luciferase activity, stimulation with PM in combination with TGF-β2 knockdown significantly increased the activity of this TGF-β signaling-specific reporter as compared to the same osmotic stimulation without TGF-β2 knockdown (*p* = 0.02). 

*COL2A1* gene expression data were confirmed on protein level. As shown in Figure 4C, PM increased COL2A1 protein levels. In addition, COL2A1 protein levels were higher at PM combined with TGF-β2 knockdown than in HACs cultured at PM alone (Figure 4D).

Since BMP signaling has been shown to be also involved in *COL2A1* expression, we next evaluated BMP-specific BRE reporter activity under the same in vitro conditions. As shown in Figure 5A, PM significantly increased BRE activity compared to the OCM control condition (*p* = 0.002). In contrast, TGF-β2 knockdown did not alter BRE activity at both PM and OCM compared to conditions without TGF-β2 knockdown. Under TGF-β2 knockdown, PM again increased BMP reporter activity as compared to OCM (*p* = 0.002, Figure 5A). 

We further evaluated the effect of an established BMP signaling inhibitor, Dorsomorphin, on *COL2A1* expression levels when TGF-β2 knockdown was combined with physosmotic stimulation. 

First, we confirmed the attenuating effect of inhibiting BMP signaling under culture conditions without TGF-β2 knockdown. BMP signaling inhibition significantly decreased *COL2A1* expression levels at both PM and OCM compared to vehicle controls, while COL2A1 expression levels at PM were still significantly elevated compared to OCM (Figure 5B). Under culture conditions with TGF-β2 knockdown, dorsomorphin again decreased *COL2A1* expression, while *COL2A1* expression was again significantly elevated at PM as compared to OCM (Figure 5B). *COL2A1* expression levels in conditions with TGF-β2-specific knockdown in combination with blocked BMP signaling were suppressed in both OCM and PM, while *COL2A1* expression remained relatively always higher in PM as compared to OCM irrespective of the TGF-β2 knockdown and blocked BMP signalling (Figure 5C). 

## 3. Discussion

In the present study, we investigated whether physiological osmolarity (PM) is able to improve expression of key cartilage ECM marker *COL2A1* in chondrocytes (HACs), through facilitating TGF-β superfamily signalling.

We reported earlier that expansion of HACs in PM (i.e., 380 mOsm) improved *COL2A1* marker gene and protein expression [10] and now reveal a correlation between improved expression of key chondrocyte marker *COL2A1* and upregulated TGF-β family member ligands (e.g., TGF-β2, -β3) in PM. Interestingly, of all three prototypic isoforms of this family, only TGF-β2 was prominently osmo-dependently regulated in our study. Of all three human isoforms, several reports support a relatively more potent role of TGF-β2 in stimulating ECM synthesis: first, expression of TGF-β2, but not TGF-β1, is reduced in cartilage of old compared to young mice, suggesting a possible chondroprotective role [24]. Secondly, only TGF-β2 is characterized by unique upstream response elements in its P1 promoter [25], suggesting its isoform specific regulation. TGF-β2 can further suppress collagenase mediated proteolytic degradation of collagen type II in human cartilage [26] and preserved the chondrocyte phenotype in physoxic high density in vitro cultures [2]. TGF-β2 also more potently induces SMAD2 phosphorylation at 10 and 250 pM as compared to TGF-β1 and TGF-β3 [16]. Furthermore, TGF-β1 and TGF-β3, but not TGF- β2, bind to endoglin [27], which is known to inhibit SMAD3 driven ECM production in chondrocytes [28]. Collectively, this prompted us to postulate that TGF-β2 signalling might trigger the increased collagen type II expression.

More than twice as much TGF-β2 protein was measured in conditioned PM compared to medium of plasma osmolarity (OCM). As TGF-β can be secreted noncovalently attached to its prodomain, requiring additional processing to be activated from latency [16], we subsequently used a functional bio-assay to investigate TGF-β activity. A twofold increase in TGF-β activity in conditioned PM was observed, confirming our gene- and protein-expression data. We next used a functional TGF-β bioassay with a SMAD3 responsive element as TGF-β2 induced SMAD3-driven transcriptional activity may cause subsequent expression of *COL2A1* in chondrocytes [18]. The endogenous TGF-β activity measured in the conditioned PM reached 50% of the activity of the exogenously added positive TGF-β control (10 ng/mL) used in the bioassay. This translates to a concentration of about 5 ng/mL active TGF-β in conditioned PM, a concentration which is sufficient for SMAD3 phosphorylation [29]. As the TGF-β bioassay is not isoform specific, we are unfortunately unable to exclude a potential contribution of other TGF-β isoforms to the observed effects.

Thus, a limitation of our study is that we cannot rule out a contribution of antagonistic TGFBRI-like pseudo-receptors, MAPKs, like p38 MAPK or ERK1/2, in our experiments [30]. These have been shown to contribute to effects of hyperosmotic stimulation in chondrocytes [30], at least at a high osmotic value (380 vs. 550 mOsm). Tew et al. further underscored the importance of post-transcriptional regulation in response to osmotic changes and, interestingly, while downstream transcriptional regulators of TGF-β signaling were induced, expression of TGF-β receptors (i.e., TGFBR2) and certain SMADs were downregulated [30]. 

Of note, our physosmotic treatment also shifted receptor expression from TGF-β type I receptor ALK1 towards relatively more canonical TGF-β type I receptor ALK5 abundance. This is in agreement with our TGF-β bioassay, since it is well established that ALK5 potentiates TGF-β-induced SMAD3-driven transcriptional activity of the *COL2A1* promotor [18]. Cartilage-specific deletion of ALK5 induces an osteoarthritis-like phenotype in mice, underscoring its chondroprotective function in cartilage [31]. A limitation of the present study is that we could not study receptor expression on protein level to answer this, due to their low expression levels [16].

A bioinformatics-driven approach suggested an involvement of TGF-β3 at high end osmotic values [30], which we cannot confirm nor rule out as we focused on TGF-β2. The latter appeared to be the most prominently regulated isoform under physiological conditions. High osmolarity is able to down-regulate not only TGFRI mRNA expression, but also TAB1, SMAD3 and BMP signaling inhibitor Smad7 [30]. Altogether, the response to extracellular osmotic values appears complex and seems to depend not only on the species, but also on cell type and its pathological state. Even the exact zonal location the chondrocytes are derived from may also influence the extent of the response to osmotic stimulation [32].

We further aimed to elucidate the role of physosmotic-driven TGF-β2 signaling on collagen expression by a ligand-specific knockdown approach. Unexpectedly, anti-TGF-β2 RNAi increased collagen type II expression upon TGF-β2 specific knockdown. We, therefore, next investigated if a counter-regulatory increase in TGF-β bioactivity might have caused the increased *COL2A1* expression and evaluated SMAD3 driven transcriptional activity. We did this in SW1353 chondrosarcoma cells as a model for human chondrocytes [23], which resulted in a similarly increased TGF-β bioactivity. Interestingly, however, the bioactivity increased even further when physosmotic treatment was combined with TGF-β2 knockdown as compared to physosmotic treatment alone. Given that SMAD3 can transcriptionally induce *COL2A1* expression [18], this may indicate a compensatory increase in TGF-β bioactivity upon TGF-β2 knockdown under physosmotic conditions. Post-transcriptional processes may also interfere with collagen expression under these conditions and may explain the increase in type II collagen protein. Earlier, SOX9 mRNA decay was shown to be influenced by high osmolarity. This led to a p38 MAPK-dependent increase in SOX9 protein and consequently enhanced COL2A1 transcription. Of note, high end osmolarity destabilized COL2A1 mRNA and reduced its overall expression [30,33]. In freshly isolated chondrocytes, osmotic stimulation thus increased SOX9, but decreased COL2A1, mRNA levels post-transcriptionally [33]. In our study, for some experiments we used SW1353 and not freshly isolated chondrocytes. Furthermore, we shifted the osmotic environment from plasma level to a (moderate) cartilage-specific physosmotic level and not from 380mOsm to high end osmolarity (>500 mOsm) as in the latter study. Together, differences in the osmotic baseline, the quality of the osmotic shift, and a different timing may at least partially explain contradictions between reported results.

Since BMP-signalling also regulates *COL2A1* expression in chondrocytes, we additionally evaluated BMP bioactivity under physosmotic conditions in combination with TGF-β2 knockdown. Cross-talk between different TGF-β superfamily members can have synergistic or antagonistic effects [34]. Physosmotic treatment significantly increased BMP responsive element (BRE) activity under both in vitro culture conditions, with or without knockdown. There was no TGF-β2 knockdown specific effect on BRE signaling, while BMP-signaling inhibitor dorsomorphin was able to attenuate *COL2A1* expression in both conditions. When combining TGF-β2 RNAi and pharmacological inhibition (i.e., dorsomorphin), *COL2A1* expression appeared to be largely BMP-dependent (Figure 5) as BMP signaling inhibition reduced *COL2A1* expression to 20% of the baseline level and 10% of the TGF-β2 knockdown control. Thus, while TGF-β superfamily members interdependently influence expression of this chondrocyte marker, physomotic conditions apparently always favor its expression independent of our means of TGF-β superfamily signalling manipulation.

The quantity of the osmotic pressure appears to be as important as its quality. To this end, recent studies confirm that using physiological salt solutions may be more appropriate than using sorbitol. Hyperosmolar potassium treatment attenuate protein production of catabolic and inflammatory OA markers in a 3D in vitro model, at values of ~490 mOsm [35]. A study of experimentally injured rat and bovine cartilage reported chondroprotection by raising osmolarity of irrigation solutions [36]. However, also here normal saline (300 mOsm) was compared to sucrose-mediated hyperosmolar saline solution (600 mOsm) [36]. We reported earlier that sodium is specifically enriched in the ECM of cartilage [10,37,38] and using NaCl to alter osmotic values appears to be more physiological than using e.g., sorbitol or sucrose solutions. Although chondrocytes in vitro do not tolerate hyperosmolar stresses above about 380 mOsm for too long [10], in vivo the situation may be different. All experiments of the present study were performed in 2D, while results might differ in 3D cultures [32], which is a limitation.

We showed that expression of several TGF-β superfamily member ligands, their receptors and target genes is altered in cultured human chondrocytes in physosmolar in vitro culture. However, the cellular responses to extracellular osmotic changes are complex [39] and both, macromolecular crowding and high ionic strength, may trigger a plethora of cell signaling cascades. High intracellular ionic strength enhances NFAT5 activity in chondrocytic cells [10,40] and more universally in mammalian cells [39]. Basically, elevating the extracellular osmolarity causes, at least temporarily, a state of molecular crowding from the presence of a total high weight per volume concentration of functionally unrelated soluble macromolecules [41]. This may indirectly affect interactions of soluble macromolecules with membranes, other structural elements or molecular chaperones.

In summary, physosmotic conditions preserve the phenotype of articular chondrocytes in vitro, a process to which TGF-β superfamily members seem to contribute. Although we could not proof a link between increased activity of this superfamily and increased *COL2A1* expression in physosmotic cultures, the osmotic component appears prevailing. The TGF-β superfamily [42] consists of more than 35 members of which we only analyzed prototypic members (i.e., TGF-βs and BMPs) of each major branch [43] in more detail. Within this setting, our rational approach further cannot exclude TGF-β mediated responses that are independent of Smad proteins [44]. Molecular crowding might affect protein structure, folding, shape, conformational stability, binding of small molecules, enzymatic activity, protein-protein interactions, protein-nucleic acid interactions, and pathological aggregation [45] and should be addressed in future studies.

## 4. Materials and Methods

### 4.1. Cartilage and Chondrocyte Isolation

Human articular cartilage was explanted from macroscopically normal areas of the femoral condyles and tibial plateau of nine patients undergoing total knee replacement surgery for osteoarthritis (OA). The Erasmus MC and the Maastricht University Medical Centre institutional policy on the use of residual human surgical material specifically states that no informed consent is needed in the case of residual surgical material. However, an approval from the institutional Medical Ethical Committee (MEC) for the use of this material is required. The MEC approved this study and assigned approval ID: MEC2004-322 and MEC08-4-028 (23 June 2008).

Cartilage was separated from the subchondral bone and cut into small pieces using a sterile surgical blade. Cartilage pieces were digested overnight at 37 °C in collagenase type II solution (300 U/mL in HEPES buffered DMEM/F12 supplemented with antibiotics (Invitrogen, Carlsbad, CA, USA), 280 mOsm) under continuous agitation [10,46,47,48]. Medium osmolarity was experimentally adjusted to 380 mOsm (physosmotic medium i.e., PM) by adding sterile NaCl as reported earlier [10]. The preparation was rinsed with 0.9% NaCl over a 70 µm cell strainer and plated in culture flasks.

### 4.2. Chondrocyte Expansion and Culturing

Briefly, primary HACs were cultured (humidified atmosphere at 37 °C, 5% CO_2_) for expansion in monolayers at a seeding density of 7500 cells/cm^2^ in culture medium consisting of DMEM/F12 (Invitrogen) with 10% fetal calf serum (FCS) (Sigma-Aldrich, St. Louis, MO, USA) and 1% antibiotic/antimycotic (Invitrogen) corresponding to their isolation osmolarity; 280 mOsm (osmotic control medium, i.e., OCM) or 380 mOsm (physosmotic medium i.e., PM). Once cells reached confluency, they were trypsinized, resuspended, and replated into 175 cm^2^ flasks. Within four weeks after harvest early passage HACs (P1, P2) were seeded in high-density monolayers (20,000 cells/cm^2^) and were cultured in OCM or PM, respectively, for an additional 7 days before analysis. Moreover, HACs were cultured for 7 days, including a 72 h serum free period in order to collect conditioned medium for TGF-β protein analyses (TGF-β2 isoform-specific ELISA and TGF-β bioassay). Experiments were performed in technical replicates from six donors.

### 4.3. RNA Expression Analysis

According to the Minimum Information for Publication of Quantitative Real-Time PCR Experiments (MIQE) guidelines [49], we subsequently aim at reporting suggested essential information: (i) the experimental design is outlined elsewhere in this section, with the numbers within each group reported as “n” per analyses, (ii) the sample dissection, processing and storage has been reported earlier by us [10], (iii) total RNA extraction, purification, and quantification thereof are also reported in much detail [10], as was (iv) the RT reaction, both using commercially available Eurogentec kits. Other essential details on the reaction conditions of the cDNA synthesis has also been previously reported [10]. With respect to (v) qPCR target information, we aimed at using the latest Entrez Gene Symbols, designed our assays using the National Center for Biotechnology Information (NCBI)’s database RefSeq sequences [50], performed NCBI’s Primer-Blast to verify specificity in silico and report NCBI Gene IDs. Real-time quantitative PCR (RT-qPCR) was performed using Mesagreen^TM^ qPCR mastermix plus for SYBR Green (Eurogentec, Maastricht, The Netherlands) and a CFX96 Real-Time PCR Detection system (Biorad, Hercules, CA, USA for amplification with the following profile: initial denaturation 10 min at 95 °C, followed by 50 cycles of amplification (15 s at 95 °C and 1 min at 60 °C), and followed by a dissociation curve. Specificity of the Primer Designer^TM^ (ThermoFisher Scientific, Waltham, MA, USA) designed and mfold-checked amplicons (> 70 < 250 bp; default secondary structure exclusion settings) were checked by agarose gel electrophoresis to exclude primer dimers and post run melting curve evaluation and assays were selected to have PCR efficiencies of >0.95. Data analysis: data were normalized to an index of stably-expressed reference genes that were pre-evaluated to be stably expressed across samples by BestKeeper software (Microsoft, Redmond, WA, USA) as was reported elsewhere by our group [2,10]. All NTCs had C_q_s ≥ 39 and GOIs with C_q_s ≥ 35 were excluded from analyses. Relative expression was calculated according to the delta-delta C_T_ method. Not earlier reported oligonucleotide sequences are listed in Table 1 and the statistical methods are reported elsewhere in this section.

### 4.4. Immunoblotting

HACs were washed with 0.9% NaCl and lysed in RIPA buffer. Extracts were sonicated on ice using the Soniprep 150 (MSE, London, UK). Insoluble material was removed by centrifugation (13,000× *g*, 4 °C). The BCA protein assay (Sigma-Aldrich) was used to determine protein concentration. Polypeptides were separated by SDS-PAGE (sodium dodecyl sulfate polyacrylamide gel electrophoresis) and transferred to nitrocellulose membranes by electroblotting. Primary antibodies (all 1:100 dilution) for immunodetection were polyclonal goat anti-COL2A1 1320–01, Southern Biotech, Birmingham, AL, USA), and mouse monoclonal anti-GAPDH (10R-1261096, Fitzgerald, MA, USA). Bound primary antibodies were detected with secondary immunoglobulins conjugated with horseradish peroxidase (Dako) and visualized by enhanced chemiluminescence (ECL). ECL signals were quantified using ImageJ 1.46f software (National Institutes of Health, Bethesda, MA, USA), and relative differences, corrected for background and housekeeper, were determined as compared to control conditions.

### 4.5. TGF-β2 ELISA

To determine TGF-β2 in the secretome, medium was collected and centrifuged at 1200 rpm for 8 min to remove cell debris. The cleared supernatant was collected and stored at –80 °C until further use. TGF-β2 concentration was quantified spectrophotometrically using an enzyme-linked immunosorbent assay (ELISA; Demeditec Diagnostics GmbH, Kiel, Germany). TGF-β2 concentrations in the samples were calculated from a standard curve according to manufacturer’s instructions.

### 4.6. TGF-β Bioassay

To verify that the secreted TGF-β as measured by ELISA is biologically active, we next employed an established luciferase assay, using the CAGA-luc Smad2/3 reporter.

Human embryonic kidney cells were seeded at 3 × 10^4^/well in 24-well plates and transiently transfected with 150 ng of the reporter construct and 75 ng of pRL-TK vector (Promega, Madison, WI, USA), an internal control for transfection efficiency, using FuGENE 6 transfection reagent (Roche Diagnostics, Basel, Switzerland). Twenty-four hours after transfection, cells were incubated for 2 h in medium containing 0.2% FCS, followed by 16 h incubation with 10 ng/mL TGF-β2 as a positive control, DMEM/F12 as a negative control or 300 μL conditioned medium (see above, ELISA). Twenty-four hours after stimulation, the firefly and Renilla luciferase activities were measured using the Dual-Luciferase Reporter Assay System (Promega, Madison, WI, USA).

### 4.7. RNAi Experiments

Primary HACs from three OA patients were cultured at OCM for two weeks and then seeded in high-density monolayers (20,000 cells/cm^2^). A TGF-β2-specific siRNA duplex (Table 1) and a scrambled siRNA control duplex were used (Eurogentec, Seraing, Belgium) [47]. HACs were transfected with siRNAs (100 nM) at OCM using HiPerFect (Qiagen, Hilden, Germany) according to the manufacturer’s protocol. One day after transfection, HACs were refreshed with either OCM or PM in the presence or absence of dorsomorphin (10 µM, Santa Cruz Biotechnologies, Santa Cruz, CA, USA). Forty-eight hours after stimulation cells were harvested for mRNA expression or protein analyses. HACs were also seeded and cultured with either DMSO (1:1000) or dorsomorphin (10 µM). One day after seeding, cells were either refreshed with OCM or PM, and 48 h later harvested for gene expression analyses. In another experiment, SW1353 cells were seeded in high-density monolayers (30,000 cells/cm^2^. Cells were transfected with the previously described CAGA-luc Smad2/3 reporter or the BRE-luc reporter using FuGene transfection reagent (Promega, Madison, WI, USA) at OCM [51]. The gaussia luciferase reporter construct was used as an internal control for transfection efficiency. One day after transfection of the reporters, the previously described TGF-β2-specific siRNA duplex and a scrambled siRNA control duplex were transfected (100 nM) at OCM using HiPerFect (Qiagen, Hilder, Germany). Five hours after siRNA transfection medium was refreshed with either OCM or PM. Forty-eight hours later, cells were processed and their luciferase activity was measured by a luciferase reporter assay system (Promega, Madison, WI, USA).

### 4.8. Statistical Analysis

Statistical analysis was performed using SPSS 13.0 software (SPSS Inc., Chicago, IL, USA). Continuous variables were tested for normality using Kolmogorov-Smirnov test and normality plots were visually assessed for skewness. All samples for gene expression analysis and TGF-β2 measurements in medium were processed and analyzed individually with replicate measurements per donor. Gene expression levels, TGF-β2 protein expression and luciferase activity between HACs cultured in PM or OCM were compared by an unpaired *t*-test. Data obtained from RNAi experiments were analyzed by a mixed linear model test. In the model, conditions with or without TGF-β2 knockdown and with or without physosmotic treatment were considered as fixed factors, while the cartilage donor was considered as a random factor. Adjustment for multiple comparisons was made by a Bonferonni’s post hoc comparison test. *p* < 0.05 was considered to indicate levels of statistically significant difference.

## Figures and Tables

**Figure 1 ijms-20-00795-f001:**
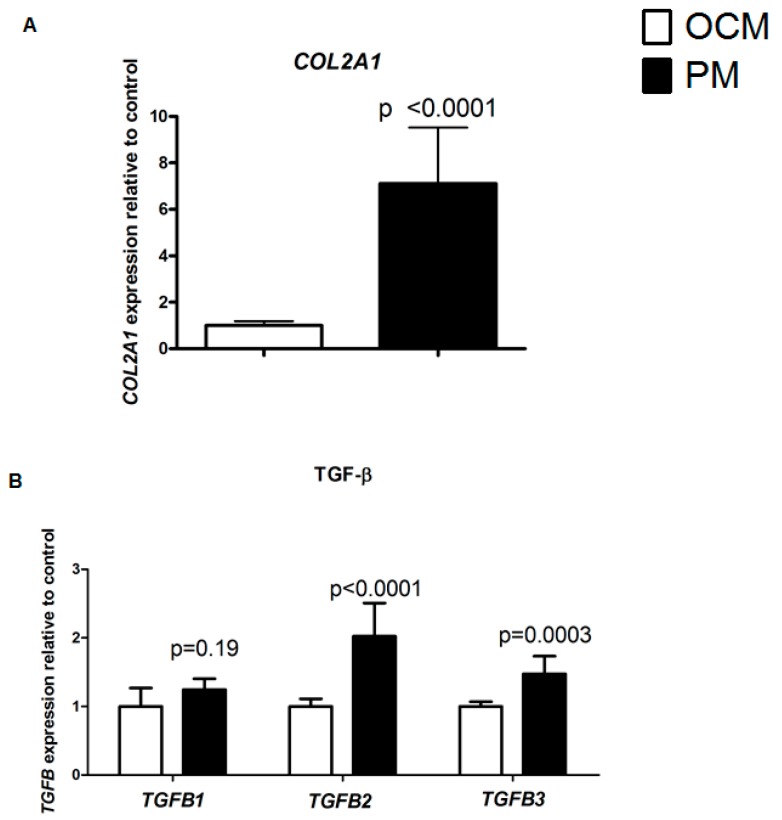
Physosmolarity-induced changes in gene expression of chondrocytes in vitro. Isolating and expanding HACs at PM for 7 days (black bars) significantly increased gene expression of *COL2A1* (**A**), as well as *TGFB2* and *TGFB3* as compared to control (OCM; white bars) (**B**). Gene expression of *TGFB1* was not significantly affected (**B**). mRNA levels were determined relatively to control OCM conditions by RT-qPCR (normalized for housekeeper expression) in HACs. Data are from six donors measured in duplicate and presented as the mean ± standard deviation.

**Figure 2 ijms-20-00795-f002:**
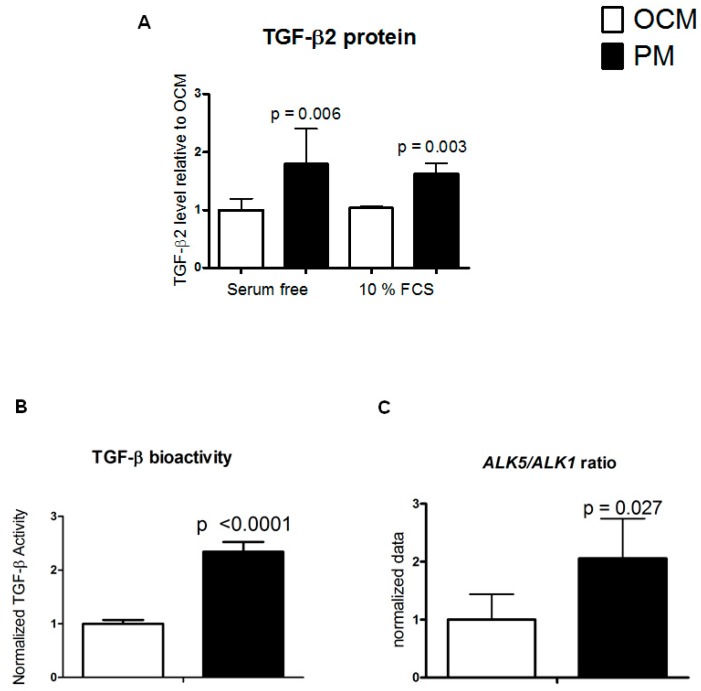
Physosmotic culture medium (PM) increases TGF-β2 secretion and TGF-β bioactivity in HACs. In HACs cultured for seven days (serum-free cultures on the left and 10% FCS cultures on the right) TGF-β2 protein levels were determined by specific TGF-β2 ELISA in PM (black bars) or OCM (white bars) (**A**). TGF-β bioactivity was determined by TGF-β bioassay in 72 h serum starved conditioned medium from PM and OCM HAC cultures (**B**). Corresponding ALK5/ALK1 gene expression ratio was determined in samples from Figure 1 (**C**). Data are from 3 donors measured in duplicate and are presented as mean ± standard deviation relative to control OCM conditions.

**Figure 3 ijms-20-00795-f003:**
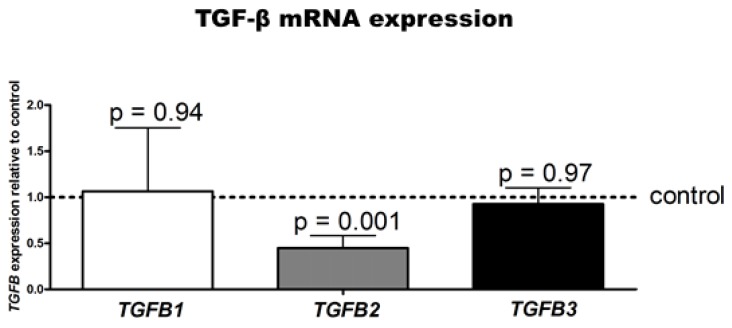
TGF-β2 RNAi efficacy and specificity in HACs. *TGFB1, TGFB2*, and *TGFB3* gene expression was determined by RT-qPCR upon TGF-β2-specific knockdown using transfected siRNAs (in OCM conditions). Data are the means ± standard deviation of duplicate measurements from three donors relative to control conditions (scrambled siRNA) and were normalized to housekeeper expression. Control condition is indicated by the dotted line (set to 1).

**Figure 4 ijms-20-00795-f004:**
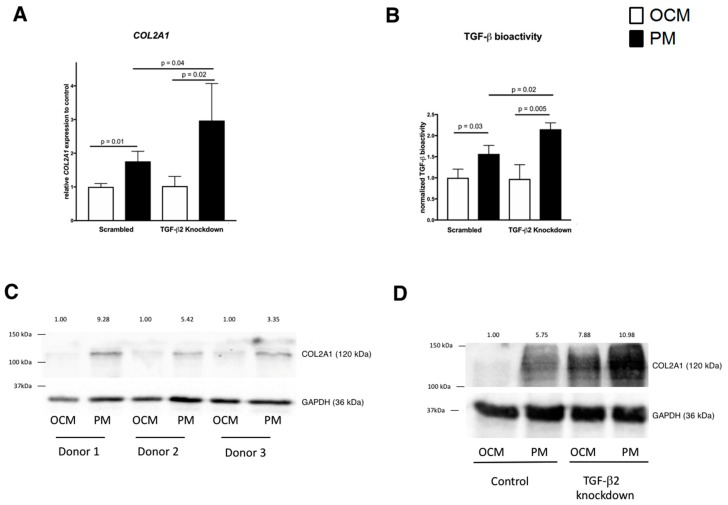
TGF-β2 RNAi combined with physosmotic treatment increases COL2A1 gene expression and TGF-β bioactivity in HACs. Expression of *COL2A1* was measured by RT-qPCR replicates (n = 3) in control conditions (**left**) and after TGF-β2 RNAi (**right**) in OCM (white bars) or PM (black bar) conditions 48 h after physosmotic stimulation (**A**). TGF-β bioactivity in media from SW1353 cultures with or without TGF-β knockdown from PM and OCM cultures. Data are normalized means ± standard deviation relative to control OCM condition (**B**). Collagen type II (COL2A1) protein expression was confirmed by immunoblotting in HACs from three donors. Molecular weight markers in kDa are shown on the left. The numbers across the top depict the relative quantity (**C**,**D**).

**Figure 5 ijms-20-00795-f005:**
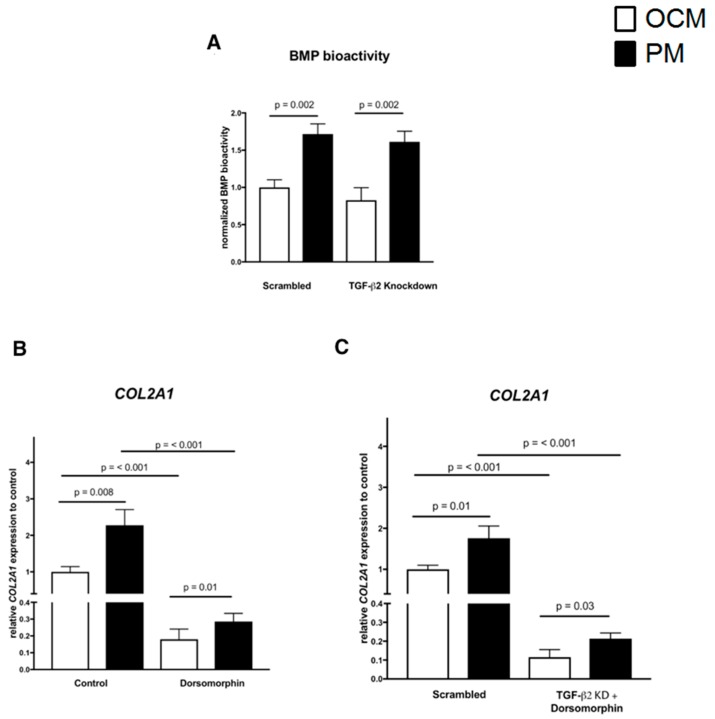
Blocking of BMP signalling decreases *COL2A1* expression of chondrocytes irrespective of osmolarity. Bioassays in SW1353 cells show a largely TGF-β2 knockdown-independent, but osmolarity-dependent induction of BMP-responsiveness. Data are the normalized means ± standard deviation (n = 3) relative to control OCM condition (**A**). Physosmolar conditions further significantly elevated *COL2A1* expression as compared to vehicle controls at OCM (**B**). Dorsomorphin (10 μM)-mediated inhibition of BMP signaling significantly suppressed *COL2A1* expression in both osmolarities (**B**), but *COL2A1* expression remained relatively higher in PM as compared to OCM irrespective of BMP signaling. Combining TGF-β2 knockdown and dorsomorphin showed the same pattern as dorsomorphin alone (**C**). Data from five donors were measured in duplicates and are presented as normalized means ± standard deviation relative to control OCM condition. Data is normalized means ± standard relative to control OCM condition (replicate measurements, three donors).

**Table 1 ijms-20-00795-t001:** Oligonucleotide sequences.

Gene Amplicon	ID T_m_	Forward/Sense (5′–3′) nt Positions	Reverse/Antisense (5′–3′) nt Positions
*COL2A1* 44 bp	1280 62.0/62.4	TGGACGATCAGGCGAAACC 3570–3588	GCTGCGGATGCTCTCAATCT 3813–3794
*TGFB1* 209 bp	7040 61.4/60.7	CTAATGGTGGAAACCCACAACG 334–355	TATCGCCAGGAATTGTTGCTG 542–522
*TGFB2* 154 bp	7042 62.3/62.9	CCATCCCGCCCACTTTCTAC 434–453	AGCTCAATCCGTTGTTCAGGC 587–567
*TGFB3* 121 bp	7043 60.9/60.3	GGAAAACACCGAGTCGGAATAC 279–300	GCGGAAAACCTTGGAGGTAAT 399–379
*ALK1* 194 bp	94 62.3/62.3	CATCGCCTCAGACATGACCTC 777–797	GTTTGCCCTGTGTACCGAAGA 970–950
*ALK5* 167 bp	7046 60.6/61.1	ACGGCGTTACAGTGTTTCTG 94–113	GCACATACAAACGGCCTATCTC 260–239
*GAPDH* *101 bp*	2597 62.0/62.9	CTGGGCTACACTGAGCACC 694–712	AAGTGGTCGTTGAGGGCAATG 794–774
*TGF-β2* *siRNA*	−	CTAATGGTGGAAACCCACAACG	TATCGCCAGGAATTGTTGCTG

Primer sequences for RT-qPCR and oligonucleotide sequences for RNAi are listed in the 5′–3′ direction. Gene, NCBI Gene Symbol; ID, NCBI Gene ID; bp, base pairs; nt positions, location of primers. Note the following newer aliases for above-listed gene symbols: ALK1 (ACVRL1), ALK5 (TGFBR1).

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
