# Peer review of "Increased TGF-β and BMP Levels and Improved Chondrocyte-Specific Marker Expression In Vitro under Cartilage-Specific Physiological Osmolarity"

_ijms, 2019, doi:10.3390/ijms20040795_

Reviewer 1 Report

The manuscript is well written and the results presented are convincing. The panel of methods applied is very useful for a deeper mechanistic understanding of chondrocyte`s response to osmolarity. The content is novel and provides useful insights into the effect of osmolarity on a very crucial signaling pathway in chondrogenesis.

For my opinion some continuative aspects could be discussed as stated below.

For the discussion: Some novel manuscripts concerning the effects of osmolarity on chondrogenic cells could be cited.

NaCl was added in the present study to adapt osmolarity. Other studies used sorbitol for mesenchymal stromal cells (Ahmadyan et al., 2018, PMID 29506631) or tested other compounds (Erndt-Marino et al., 2017, PMID28992763). Why did the authors decide to use NaCl and not sorbitol? It could be that different agents might evoke a differing response in chondrocyte. At which level could hyperosmolarity be expected? E.g. Eltawil et al., 2018, PMID 29156946), at which lever becomes it detrimental?

Since the experiments were performed in monolayer culture, it could be discussed that the influence of osmolarity might differ in 3D culture (e.g. Takada and Mizuno, 2018 PMID:29783650)?

Indeed, type II collagen is a key cartilage marker, but how about other markers (e.g. aggrecan is most important for reversible water binding in the ECM or more upstream Sox9). The authors could discuss other markers/relevant proteins regulated by TGFβ2,3?

Minor issues

Abstract

„dorsomorphin“ please explain with the first mention that it is a BMP inhibitor (line 28 vs. 32).

Line 30: „Physiological cartilage osmolarity“, „physosmolarity“ could already be introduced here and used e.g. line 32

Line 37 „physomotic“ write „physosmotic“

Introduction

Line 70 the bracket has to be closed „(TGF-β“

Results

Figure 1: please write in the legend at which time point the gene expression was determined

Line 120 insert a blank before the citation

Line 159 „We repeated the experiment in the established chondrosarcoma cell line SW1353“

Why was this cell line used for confirmation – chondrosarcoma cells could be less responsive compared with primary HACs?

Fig. 4C (Legend) why is donor description doubled? Insert a point behind „(B)“.

„KdA“, better to write „kDa“ for kilodaltons?

Line 219 and 222: remove the surplus blank/point. Line 240: insert a blank before citation.

Line 243: „which is notoriously difficult due to their low expression levels.“ Please support this statement by a reference.

Line 268: „20%“ and „10 %“, please use either a blank or not. Check this throughout the whole manuscript. Decide also consequently, whether you abbreviate or not the units „d“ vs „day“ etc.

Line 379: correct „refresheid“

Line 380: Add a blank: „WI,USA“

I think the reference list is not in the correct formate of the journal

Author Response

 Reviewer 1

Comments and Suggestions for Authors

The manuscript is well written and the results presented are convincing. The panel of methods applied is very useful for a deeper mechanistic understanding of chondrocyte`s response to osmolarity. The content is novel and provides useful insights into the effect of osmolarity on a very crucial signaling pathway in chondrogenesis.

 Response: Thank you for your positive evaluation of our manuscript.

For my opinion some continuative aspects could be discussed as stated below.

For the discussion: Some novel manuscripts concerning the effects of osmolarity on chondrogenic cells could be cited.

NaCl was added in the present study to adapt osmolarity. Other studies used sorbitol for mesenchymal stromal cells (Ahmadyan et al., 2018,PMID 29506631) or tested other compounds (Erndt-Marino et al., 2017, PMID28992763). Why did the authors decide to use NaCl and not sorbitol? It could be that different agents might evoke a differing response in chondrocyte. At which level could hyperosmolarity be expected? E.g. Eltawil et al., 2018, PMID 29156946), at which lever becomes it detrimental?

Response: This is correct: osmolarity could be controlled by different means and other groups used sorbitol or related sugars to change it. We used NaCl to do so for several reasons: first, sodium is specifically very enriched in the cartilage extracellular matrix as compared to plasma. For chondrocytes, sodium is thus a rather physiological means to alter osmolarity, while e.g. sorbitol is probably not. Secondly, prior to our first study (van der Windt et al. 2010, PMID:20492652) we compared selected sugar derivatives to NaCl and found that their effects on gene expression of selected chondrocytic markers were largely identical. Of note, we also showed earlier that important chondrocyte markers were responsive to elevated osmolarity, induced by NaCl, and demonstrated that these effects were majorly mediated by NFAT5 (e.g., van der Windt et al. 2012, PMID: 22231955 and Caron et al. 2013, PMID:23219947). Also, we always aimed at using similar baseline osmolarities in all our previous studies, being close to 290 mOsm/kg, to be able to compare results. This also holds for the present study. As some cell lines require slightly different medium than (primary) chondrocytes, a marge of +/- 10 mOsm/kg was allowed for baselines. Importantly, the relative increase in osmolarity in our experimental conditions always occurred in increments of +100mOsm/kg. Thus, by adding NaCl to culture media, no detrimental effects are expected to occur below about 450mOsm/kg, as reported in our earlier studies by van der Windt and Caron.

Using tracked changes, we now added a short passage to our Discussion (in red).

Since the experiments were performed in monolayer culture, it could be discussed that the influence of osmolarity might differ in 3D culture (e.g. Takada and Mizuno, 2018 PMID:29783650)?

 Response: Thank you for this suggestion, which is correct. In our Discussion, we now added “A limitation of the present study is that all experiments were performed in 2D, while results might differ in 3D cultures” (Ref. #30: Takada and Mizuno, 2018 PMID:29783650) to the manuscript.

Indeed, type II collagen is a key cartilage marker, but how about other markers (e.g. aggrecan is most important for reversible water binding in the ECM or more upstream Sox9). The authors could discuss other markers/relevant proteins regulated by TGFβ2,3?

Response: We agree that expression of aggrecan and Sox-9, as two other key cartilage markers, could have been shown in this study, too. However, we have routinely looked into many more markers, including these two, in similar studies in the past. In general, trends in fold-change of gene expression of these markers were similar to those of COL2A1 (e.g., van der Windt et al. 2012, PMID: 22231955 and Caron et al. 2013, PMID:23219947). Therefore, we restricted ourselves to COL2A1 in this study as proper expression of this gene is usually the problem in OA. Aggrecan is synthesized, but cannot be “anchored” to the ECM anymore, due to a missing network of proper collagen type II.

For reviewing, we now added new data for which we used cultured human articular knee chondrocytes from two OA patients in a lentiviral TGF-b2 knockdown approach with the same layout to also screen expression of aggrecan and Sox-9 upon PM stimulation:

Figure 1 (attached): Upon confirming TGF-b2 RNAi specificity first (A), by screening mRNA expression of all three TGF-b isoforms 48 hours after transduction, we evaluated gene expression of the suggested markers (B-D) and saw indeed a similar trend for all three “markers”. Data are presented as mean ± standard deviation of replicate RT-qPCRs (n= two donors). OCM, osmotic control medium, PM, physosmotic medium.

Minor issues

Abstract

„dorsomorphin“ please explain with the first mention that it is a BMP inhibitor (line 28 vs. 32).

Line 30: „Physiological cartilage osmolarity“, „physosmolarity“ could already be introduced here and used e.g. line 32

Line 37 „physomotic“ write „physosmotic“

Introduction

Line 70 the bracket has to be closed „(TGF-β“

Response: We performed the above-mentioned minor corrections in the revised version, now.

Results

Figure 1: please write in the legend at which time point the gene expression was determined

Line 120 insert a blank before the citation

Response: We also corrected this in the revised version, now.

Line 159 „We repeated the experiment in the established chondrosarcoma cell line SW1353“

Why was this cell line used for confirmation – chondrosarcoma cells could be less responsive compared with primary HACs?

Response: Thank you for this question. There is a plethora of studies indicating that SW1353 is a good in vitro chondrocyte model (e.g., Ouyang P., PMID: 9618287; Gebauer et al. PMID:15950496). On the other hand, availability of primary human articular chondrocytes is often limited and these cells are well known to have limited transfection efficiencies.

Fig. 4C (Legend) why is donor description doubled? Insert a point behind „(B)“.

Response: Thank you. The suggested corrections were made.

„KdA“, better to write „kDa“ for kilodaltons?

 Response: Sorry, has been corrected, now.

 Line 219 and 222: remove the surplus blank/point. Line 240: insert a blank before citation.

 Response: Has been corrected.

Line 243: „which is notoriously difficult due to their low expression levels.“ Please support this statement by a reference.

Response: We added Ref. Parker WL et al., now.

line 268: „20%“ and „10 %“, please use either a blank or not. Check this throughout the whole manuscript. Decide also consequently, whether you abbreviate or not the units „d“ vs „day“ etc.

Line 379: correct „refresheid“

Line 380: Add a blank: „WI,USA“

Response: Thank you, corrections were made.

I think the reference list is not in the correct formate of the journal

Response: Thanks for pointing this out. We changed the style and will also check with the Editorial Office.

Reviewer 2 Report

The authors investigated that TGF-β superfamily signaling may drive expression of COL2A1 under physiological osmolarity culture conditions. They demonstrated cartilage physiological osmolarity-induced COL2A1 mRNA expression is regulated by TGF-β superfamily members interdependently and that physomotic conditions favor the expression of COL2A1 independent of the manipulation. This manuscript is clearly written with sufficient data to claim the authors' findings. Only, I recommend them to discuss and illustrate (or summarize) how the physosmotic treatment increases TGF-β family expression and COL2A1 mRNA as a sequential event.

Author Response

Reviewer 2

The authors investigated that TGF-β superfamily signaling may drive expression of COL2A1 under physiological osmolarity culture conditions. They demonstrated cartilage physiological osmolarity-induced COL2A1 mRNA expression is regulated by TGF-β superfamily members interdependently and that physomotic conditions favor the expression of COL2A1 independent of the manipulation. This manuscript is clearly written with sufficient data to claim the authors' findings. Only, I recommend them to discuss and illustrate (or summarize) how the physosmotic treatment increases TGF-β family expression and COL2A1 mRNA as a sequential event.

Response: We appreciate your positive evaluation of our manuscript. In response to your suggestion, in our revised Discussion (esp. L 267 following), we illustrated putative mechanistic explanations of how physosmotic stimulation may increase COL2A1 expression through improved TGF-b superfamily signaling verbally, now.

Reviewer 3 Report

The authors use monolayer cultures of human OA articular chondrocytes. They show that increasing osmolarity from standard culture medium levels to physiological levels present in articular cartilage leads to increase of TGFb mRNA and protein levels and bioactivity. Also BMP bioactivity increases. In the same procedure the expression of Col2A1 mRNA and protein increases. However, the experiments performed do not prove a causal relationship between increased activity of the TGFb superfamily and the increased COL2A1 levels. On the contrary: Blocking of TGFb2, the most induced isoform in physosmotic culture conditions, leads to increased COL2A1 mRNA and protein (Fig4). In osmotic control medium blocking of TGFb2 has no effect on TGFb activity, but also here COL2A1 protein increases (Fig 4). This is not properly addressed in the discussion part. Also the blocking of BMP with dorsomorphin in both PM and OCM conditions provides no proof for involvement of the TGFb superfamily in the underlying mechanisms of osmolarity-induced shift in type II collagen expression. Possibly, at physosmotic conditions (or at increased sodium concentrations) the metabolic activity of chondrocytes increases without specific involvement of the TGFb superfamily.

For these reasons the title of the manuscript and the conclusions part of the abstract should be changed, because of two incorrect claims.

1)OA chondrocytes in monolayer are studied, not cartilage.

2)No proof is provided for a causal relationship between increased activity of the TGFb superfamily and the increased COL2A1 levels.

In all figures it is unclear how many times experiments were repeated and how many different donors were used.

Line 37: physomotic instead of physosmotic.

Line 136: in samples from figure 1. Why is n=6 here and n+12 there?

Line 168: in the results of figure 4 it is not mentioned that TGFb2 knockdown in OCM conditions did not change TGFb bioactivity, but did increase type II collagen protein. Also in the discussion part this was not addressed.

Lines 186-192 are part of the legends of figure 4

Lines 200 and 202: the beta is lacking

Lines 200-203:"slightly further" this difference is not statistically significant, so you cannot speak of a TGF-b2-specific contribution.

Lines 301-303: After 4 weeks of expansion and 1 week at OCM or PM, can one still speak of "primary chondrocytes (line 275)?

Author Response

Reviewer 3

The authors use monolayer cultures of human OA articular chondrocytes. They show that increasing osmolarity from standard culture medium levels to physiological levels present in articular cartilage leads to increase of TGFb mRNA and protein levels and bioactivity. Also BMP bioactivity increases. In the same procedure the expression of Col2A1 mRNA and protein increases. However, the experiments performed do not prove a causal relationship between increased activity of the TGFb superfamily and the increased COL2A1 levels. On the contrary: Blocking of TGFb2, the most induced isoform in physosmotic culture conditions, leads to increased COL2A1 mRNA and protein (Fig4). In osmotic control medium blocking of TGFb2 has no effect on TGFb activity, but also here COL2A1 protein increases (Fig 4). This is not properly addressed in the discussion part. Also the blocking of BMP with dorsomorphin in both PM and OCM conditions provides no proof for involvement of the TGFb superfamily in the underlying mechanisms of osmolarity-induced shift in type II collagen expression. Possibly, at physosmotic conditions (or at increased sodium concentrations) the metabolic activity of chondrocytes increases without specific involvement of the TGFb superfamily.

For these reasons the title of the manuscript and the conclusions part of the abstract should be changed, because of two incorrect claims.

1) OA chondrocytes in monolayer are studied, not cartilage.

Response: Thank you, you are right. We now added a section in our revised Discussion to address this limitation of our study using tracked changes (in red). We further rewrote parts of our Discussion and adjusted our Conclusions accordingly. Please see tracked changes in body text (in red).

2) No proof is provided for a causal relationship between increased activity of the TGFb superfamily and the increased COL2A1 levels.

Response: We appreciate this concern. We consistently found that manipulation of TGF-β superfamily signaling with our means caused a relative larger change in the marker gene expression than osmotic stimulation alone. This strongly suggests that TGF-β superfamily members participate in the observed effects. However, as we could not clearly proof the magnitude of the respective contribution of a specific factor of this large superfamily in the present study, we now modified our Conclusions and the Title in response to this concern.

The revised Conclusions now read as follows: Our data suggest a role for TGF-β superfamily member signaling in physosmolarity-induced mRNA expression of collagen type II. As physosmotic conditions favor the expression of COL2A1 independent of our manipulations, contribution of other metabolic, post-transcriptional or epigenetic factors cannot be excluded in the underlying complex and interdependent regulation of marker gene expression. Dissecting these molecular mechanisms holds potential to further improve future cell-based chondral repair strategies.

We changed the Title of our study into: TGF-β superfamily signaling and improved chondrocyte-specific marker expression in vitro under cartilage-specific physiological osmolarity.

However, the experiments performed do not prove a causal relationship between increased activity of the TGFb superfamily and the increased COL2A1 levels. On the contrary: Blocking of TGFb2, the most induced isoform in physosmotic culture conditions, leads to increased COL2A1 mRNA and protein (Fig4). In osmotic control medium blocking of TGFb2 has no effect on TGFb activity, but also here COL2A1 protein increases (Fig 4). This is not properly addressed in the discussion part. Also the blocking of BMP with dorsomorphin

 Response: You are right and we therefore now revised our Discussion (L 267-) accordingly, also addressing other not before-mentioned putatively contributing factor and regulatory feedback loops. Please see tracked changes in body text (in red). Please note that in response to Reviewer 1 we now added new data for which we used cultured human articular knee chondrocytes from two OA patients and used a lentiviral TGF-b2 knockdown approach. The new data also hint towards the importance of timing and cell source as has been addressed elsewhere, now.

 In all figures it is unclear how many times experiments were repeated and how many different donors were used.

 Response: Thank you. We apologize for this lack of information and now added this to all figures. Changes are performed in red and as tracked changes.

 Line 37: physomotic instead of physosmotic.

Response: Thank you. This has now been corrected.

Line 136: in samples from figure 1. Why is n=6 here and n+12 there?

Response: For some experiments, we could not use chondrocytes from six donors in replicates, but only from three, as we were not able to harvest sufficient cell numbers from the other donors.  due to the limited availability of chondrocytes. In these cases, we got less replicates for statistics.

 Line 168: in the results of figure 4 it is not mentioned that TGFb2 knockdown in OCM conditions did not change TGFb bioactivity, but did increase type II collagen protein. Also in the discussion part this was not addressed.

 Response: Thank you for this comment. We now addressed this in our revised Discussion, referring to, amongst others, data from Tew and colleagues. Line 267 following now reads as follows: “We further aimed to elucidate the role of physosmotic-driven TGF-β2 signaling on collagen expression by a ligand-specific knockdown approach. Unexpectedly, anti-TGF-β2 RNAi increased collagen type II expression upon TGF-β2 specific knockdown. We, therefore, next investigated if a counter-regulatory increase in TGF-β bioactivity might have caused the increased COL2A1 expression and evaluated SMAD3 driven transcriptional activity. We did this in SW1353 chondrosarcoma cells as a model for human chondrocytes (23), which resulted in a similarly increased TGF-β bioactivity. Interestingly, however, the bioactivity increased even further when physosmotic treatment was combined with TGF-β2 knockdown as compared to physosmotic treatment alone. Given that SMAD3 can transcriptionally induce COL2A1 expression (18), this may indicate a compensatory increase in TGF-β bioactivity upon TGF-β2 knockdown under physosmotic conditions. Post-transcriptional processes may also interfere with collagen expression under these conditions and may explain the increase in type II collagen protein. Earlier, SOX9 mRNA decay was shown to be influenced by high osmolarity. This led to a p38 MAPK-dependent increase in SOX9 protein and consequently enhanced COL2A1 transcription. Of note, high end osmolarity destabilized COL2A1 mRNA and reduced its overall expression (30, 33). In freshly isolated chondrocytes, osmotic stimulation thus increased SOX9, but decreased COL2A1, mRNA levels post-transcriptionally (33). In our study, for some experiments we used SW1353 and not freshly isolated chondrocytes. Furthermore, we shifted the osmotic environment from plasma level to a (moderate) physosmotic osmolarity and not from 380mOsm to high end osmolarity (>500mOsm) as in the latter study. Together, differences in the osmotic baseline, the quality of the osmotic shift, and a different timing may at least partially explain contradictions between reported results.

 Lines 186-192 are part of the legends of figure 4

 Response: Yes. We have formatted this accordingly.

 Lines 200 and 202: the beta is lacking

Response: Thanks a lot. We corrected that.

Lines 200-203:"slightly further" this difference is not statistically significant, so you cannot speak of a TGF-b2-specific contribution.

Response: We agree and changed this in the corresponding sections of the revised version in the legend of figure 5.

Lines 301-303: After 4 weeks of expansion and 1 week at OCM or PM, can one still speak of "primary chondrocytes (line 275)?

Response: Thank you for raising this point, but we believe you actual can. During in vitro culture chondrocytes tend to de-differentiate, which is well accepted, and cells are probably not “undifferentiated” anymore despite the high seeding densities used in our approach. However, “primary chondrocytes” is used by us to more clearly differentiate the cells from cell lines. We, however, tried to phrase this more clearly, now, in the revised paper.

Round  2

Reviewer 3 Report

The authors found increased levels of BMP and TGF-β and type II collagen at physiological osmolarity, but did not unravel underlying intracellular mechanisms and did not look at intracellular signaling. Therefore, the title is still misleading. To my opinion a more correct title would be: "Increased TGF-β and BMP levels and improved chondrocyte-specific marker expression in vitro under cartilage-specific physiological osmolarity". ("levels" could also be exchanged for "activity".)

-Line 197: This sentence was corrected , but the original "increased" was ok. In line with my earlier remarks, to my opinion this sentence should be  "Blocking of BMP decreases collagen expression  of chondrocytes, irrespective of osmolarity".

-lines 208-211: This tekst was clearly meant for the legends of figure 5. I would suggest that under C) you state:"combining TGF-β2 knockdown and dorsomorphin showed the same pattern as dorsomorphin alone". Then the tekst in lines 208-211 could then be placed in the "results"part of the manuscript, at page 5.

-Lines 296-298: You cannot compare these 10% and 20%, because 5b and 5c are clearly two different experiments. In these experiments no extra effect of TGF-β2 knockdown can be claimed or denied.

-line 326-327: "While our data further link SMAD2/3 and SMAD 1/5/8 signaling pathways to increased COL2A1 expression..."

Here we go again! BMP and TGFβ clearly are important for type II collagen expression, but you found no proof for their role in the processes caused by the shift in osmolarity! I propose the following wording: "Although there might be a link between inceased activity of this superfamily and increased COL2A1 expression in physosmotic cultures,...".

Author Response

Dear sir/madam,

Thank you for your valuable suggestions you have made. Please find attached a Word document providing a point-by-point response to your comments. 
